# Comparative Benchmarking of Optical Genome Mapping and Chromosomal Microarray Reveals High Technological Concordance in CNV Identification and Additional Structural Variant Refinement

**DOI:** 10.3390/genes14101868

**Published:** 2023-09-26

**Authors:** Hayk Barseghyan, Andy Wing Chun Pang, Benjamin Clifford, Moises A. Serrano, Alka Chaubey, Alex R. Hastie

**Affiliations:** 1Bionano, San Diego, CA 92121, USA; haykbarseghyan@ucla.edu (H.B.); apang@bionano.com (A.W.C.P.); bclifford@bionano.com (B.C.); achaubey@bionano.com (A.C.); 2Center for Genetic Medicine Research, Children’s National Hospital, Washington, DC 20010, USA; 3Genomics and Precision Medicine, School of Medicine and Health Sciences, George Washington University, Washington, DC 20037, USA; 4Bionano Laboratories, Salt Lake City, UT 84109, USA; mserrano@bionano.com

**Keywords:** optical genome mapping, OGM, Saphyr, chromosomal microarray, CMA, SVs, CNVs, aneuploidy, triploidy, absence of heterozygosity, AOH

## Abstract

The recommended practice for individuals suspected of a genetic etiology for disorders including unexplained developmental delay/intellectual disability (DD/ID), autism spectrum disorders (ASD), and multiple congenital anomalies (MCA) involves a genetic testing workflow including chromosomal microarray (CMA), Fragile-X testing, karyotype analysis, and/or sequencing-based gene panels. Since genomic imbalances are often found to be causative, CMA is recommended as first tier testing for many indications. Optical genome mapping (OGM) is an emerging next generation cytogenomic technique that can detect not only copy number variants (CNVs), triploidy and absence of heterozygosity (AOH) like CMA, but can also define the location of duplications, and detect other structural variants (SVs), including balanced rearrangements and repeat expansions/contractions. This study compares OGM to CMA for clinically reported genomic variants, some of these samples also have structural characterization by fluorescence in situ hybridization (FISH). OGM was performed on IRB approved, de-identified specimens from 55 individuals with genomic abnormalities previously identified by CMA (61 clinically reported abnormalities). SVs identified by OGM were filtered by a control database to remove polymorphic variants and against an established gene list to prioritize clinically relevant findings before comparing with CMA and FISH results. OGM results showed 100% concordance with CMA findings for pathogenic variants and 98% concordant for all pathogenic/likely pathogenic/variants of uncertain significance (VUS), while also providing additional insight into the genomic structure of abnormalities that CMA was unable to provide. OGM demonstrates equivalent performance to CMA for CNV and AOH detection, enhanced by its ability to determine the structure of the genome. This work adds to an increasing body of evidence on the analytical validity and ability to detect clinically relevant abnormalities identified by CMA. Moreover, OGM identifies translocations, structures of duplications and complex CNVs intractable by CMA, yielding additional clinical utility.

## 1. Introduction

Since the completion of the human genome project, the development of molecular technologies has helped many research investigators around the world to uncover the underlying genetic causes for hundreds of disorders [1]. Compared with traditional cytogenetic technologies such as karyotyping and fluorescence in situ hybridization (FISH), novel high throughput genomic methods such as chromosomal microarrays (CMA) brought previously unseen scalability, increased resolution, and sensitivity for copy number variant (CNV) detection at a genome-wide scale. Because of these advantages, CMA has increased overall diagnostic yield to 15–20% when compared to approximately 5% using karyotyping [2]. Therefore, CMA has been recommended as a first-tier clinical diagnostic tool for individuals with developmental disabilities or congenital anomalies [3,4,5,6]. With the improved ability to detect CNVs, American College of Medical Genetics and Genomics (ACMG) and the Clinical Genome Resource (ClinGen) recently published guidelines for interpretation and reporting of CNVs in the constitutional setting [7].

While detection of CNVs by CMA has significantly improved the detection rate of chromosomal anomalies, tools for routine identification of balanced genomic rearrangements remains elusive. Additionally, the occurance of balanced structural variations (SVs) and any adverse phenotypic impact they might have, cannot be elucidated by current CMA technology [8]. Typically, karyotype is needed to identify large SV when there is no established family history, and FISH can only be used to characterize an SV when there is a known or suspected aberration that can be targeted using locus-specific probes. Identification of genomic aberrations and knowledge of the underlying genomic structure of chromosomal alterations may provide crucial information to determine recurrence risks. For example, a copy number gain identified by CMA may potentially disrupt a gene with phenotypic consequences depending on the direction and chromosomal position of that duplication.

Optical genome mapping (OGM), a completely orthogonal approach to genomics, has emerged as a high-throughput cytogenomic method that can identify all classes of SVs including insertions, deletions, duplications, inversions, translocations as well as triploidy, repeat expansions/contractions and absence of heterozygosity (AOH) [9,10,11,12,13]. The ability to detect such a wide range of balanced and unbalanced SVs and aneuploidies is facilitated by fluorescent labeling of specific genome-wide six-nucleotide sequence motifs in ultra-long DNA molecules and assembled into haplotype resolved consensus genome maps for variant calling. OGM has been used for the identification of SVs associated with constitutional disorders [14,15,16,17,18] as well as cancers [19,20,21]. These studies show that, compared to current cytogenetic methodologies, OGM is more sensitive for SV identification, with similar or better turnaround times and can be more cost effective with a simpler workflow. Here, SV detection was benchmarked by OGM against a set of clinically reported variants identified by CMA in a series of 55 samples. The results demonstrate 98% concordance using OGM with clinically reported variants identified by CMA. Also, OGM provides better characterization of the genomic architecture of SVs that would otherwise need to be elucidated by additional methods such as FISH or karyotype.

## 2. Materials and Methods

### 2.1. Cohort Composition

An IRB approved (IRB-20216077), de-identified cohort of peripheral blood specimens or cell lines (*N* = 55) from individuals referred for chromosomal analysis by CMA were subjected to OGM in this study. The cohort represents a diverse set of phenotypic indications and types of chromosomal aberrations (Appendix A). Some abnormal CMA results were subsequently assessed with FISH testing to confirm or further characterize the chromosomal structure of the aberration. The cohort was selected to include the following aberrations identified by CMA: (1) terminal and interstitial deletions and duplications; (2) whole chromosome gains and losses, including mosaic aneuploidies (e.g., trisomy, monosomy, and triploidy); (3) complex CNVs structurally refined by FISH (e.g., unbalanced translocations and insertions); and (4) AOH segments. CMA was performed using Agilent SurePrint G3 Custom CGH + SNP 4 × 180 k array with a customized protocol according to CLIA/CAP certified laboratory procedures.

### 2.2. Optical Genome Mapping

Ultra-high molecular weight (UHMW) DNA was extracted from white blood cells or cultured cells following the manufacturer’s protocols (Bionano, San Diego, CA, USA). The cells were digested with Proteinase K and RNase A. DNA was precipitated with isopropanol, bound with a nanobind magnetic disk, and washed. UHMW DNA was resuspended in the elution buffer and quantified with Qubit double-stranded DNA assay kits (Thermo Fisher Scientific, Waltham, MA, USA).

DNA labeling was performed following the manufacturer’s protocols (Bionano, USA). Direct Labeling Enzyme 1 (DLE-1) reactions were carried out using 750 ng of purified UHMW DNA. The fluorescently labeled DNA molecules were loaded on flowcells and imaged sequentially across nanochannels on a Saphyr instrument. An effective genome coverage of ~100× was achieved for all tested samples. Sample run quality thresholds were set to meet the following QC metrics: label density of ~15/100 kbp; filtered molecules N50 (≥150 kbp) ≥ 230 kbp; map rate ≥ 70%.

### 2.3. Data Analysis

The proprietary OGM-specific software—Bionano Access and Solve (versions 1.6/1.7 and 3.6/3.7, respectively), were used for data processing. Specifically, two analytical pipelines were used for variant identification: de novo assembly and fractional copy number analysis (also referred to as CNV pipeline). De novo assembly was performed using Bionano’s custom assembler software program based on the Overlap-Layout-Consensus paradigm. Pairwise comparison of all DNA molecules was performed to generate the initial consensus genome maps (*.cmap). Genome maps were further refined and extended with best matching molecules. SVs were identified based on the alignment profiles between the de novo assembled genome maps and the Human Genome Reference Consortium GRCh37 or GRCh38 assembly. If the assembled map did not align contiguously to the reference, but instead were punctuated by internal alignment gaps (outlier) or end alignment gaps (endoutlier), then a putative SV was identified. Fractional copy number analyses were performed from the alignment of molecules and labels against GRCh37/38 (alignmolvref). The raw label coverage of the samples was normalized against relative coverage from normal human controls, segmented, and the baseline CN state was estimated from calculating the mode of coverage of all labels. If chromosome Y molecules were present, baseline coverage in sex chromosomes was halved. With a baseline estimated, CN states of segmented genomic intervals were assessed for significant increase/decrease from the baseline. Corresponding copy number gains and losses were exported. Certain SV and CN calls were masked, if occurring in GRC37 regions found to be in high variance (gaps, segmental duplications, etc.). Both de novo and CNV pipelines could have an overlap in duplication and deletion calling; however, because the de novo pipeline utilizes molecules to build contigs, the resultant label locations aligned to the reference genome possess higher precision than a segment with a change in coverage depth (fractional copy number analysis). Hence, the SVs identified by the de novo pipeline were prioritized.

After filtering SV calls for high-quality, informative sites, absence of heterozygosity (AOH) events were called using a Hidden Markov Model (HMM) that models the spatial dependence between neighboring SVs of a given zygosity. Model parameters were previously estimated by fitting the model to a simulated dataset, which was generated by splicing together SV calling datasets from 153 controls and 4 haploid samples, where regions derived from haploid genomes represented AOH events. In addition, variant allele fraction (VAF) of SVs in the samples was calculated based on effective genome coverage at the SV loci. With the availability of variant allele fractions, one can infer triploidy by visual inspection. In a typical diploid sample, VAF clusters around 0.5 (heterozygous ALT-REF) and 1.0 (homozygous ALT) (Appendix A), whereas in a triploid sample, variant fraction is clustered into three groups, 0.33, 0.67 and 1.0 (Figure 1A).

Variants were initially filtered based on SV/CNV quality metrics, masking regions of the genome that are difficult to align (e.g., centromeres, telomeres, reference gaps), SV call frequency and CNV size. Briefly, all SVs and CNVs were first filtered with the recommended confidence cutoff values. Second, SV frequency in Bionano Genomics’ control database, consisting of 297 normal healthy control samples, was used to filter out common variants (>1% population frequency). Third, copy number gains/losses below 500 kbp and insertions/deletions below 500 bp were filtered out. The filtered outputs were exported into a working table for further review. All variants were assessed and classified using ACMG standards and guidelines for CNV assessment [7].

## 3. Results

### 3.1. Concordance

Of the 61 clinically significant structural variants present in 55 samples, 60 were reproduced by OGM, providing a 98% concordance between CMA and OGM (Table 1). Additionally, of the 46 reported pathogenic variants in 36 samples, all were also identified by OGM, leading to a 100% concordance for SVs that were interpreted as pathogenic following CMA.

### 3.2. Detection of Whole Chromosome Copy Gains and Losses and Copy Neutral Events

Three cases of triploidy analyzed by OGM demonstrated a copy number gain of all chromosomes. Similar to the copy number data in CMA, triploidy must be inferred due to normalization of the genome to a diploid status. With OGM, the inference is achieved by variant allele fraction calculations that cluster into three groups (see methods). Figure 1A shows an example of a triploid genome identified by OGM, Bionano Access software (v.1.7) displays lines to help visualize the heterozygous groups of VAF modes (0.33, 0.66), whereas in normal cases, the mode would be at 0.5.

A total of five cases with whole chromosome aneuploidy were evaluated using OGM: two with trisomy 13 and one each with trisomy 21, monosomy X and mosaic Y loss (Appendix A). OGM successfully identified all five aneuploidies. Notably, one trisomy 13 case identified by OGM and CMA was determined by metaphase FISH to be due to a Robertsonian translocation (Appendix A). Figure 1B shows two examples of aneuploidies identified by OGM: trisomy 13 and monosomy X. Figure 1C shows detection of a mosaic loss of the Y chromosome. Detection of absence of heterozygosity (AOH) in copy neutral genomic regions is an important consideration research for the identification of potential uniparental isodisomy and/or increased risk for autosomal recessive disease via identity by descent, thus possibly aiding in disease classification [22]. We investigated four cases with AOH and the results were concordant with those identified by CMA for events ≥ 25 Mbp in size. Smaller AOH events are currently below the detection limit of OGM. Figure 1D shows an example of uniparental disomy (UPD) where most of the SVs were homozygous for chromosome (Chr 8).

### 3.3. Microdeletion/Duplication Syndromes

OGM successfully identified all microdeletions/duplications that were identified by CMA in this cohort (Appendix A). For most of the deletions and duplications, both methods predicted similar sizes and breakpoints within the limitations of each technology. The majority of variance between size calls by the two platforms were related to the presence of low copy repeats (LCRs) flanking the CNVs (Appendix A, Figure 2). Specifically, CMA has limited coverage in repetitive regions, while OGM has long molecule contiguity spanning the LCRs.

The microdeletions/duplications shown in Figure 2 provide examples of how OGM accurately identifies these types of CNVs. For example, a 1.9 Mb deletion of 7q11.23 associated with Williams–Beuren syndrome was identified by both the CNV and de novo assembly pipelines (Figure 2A, Sample 8). The SV size differences between CMA and OGM can be attributed to the localization of the breakpoints in LCR regions, where CMA probe localization or reference assembly quality is suboptimal. Similarly, OGM identified a 1.9 Mbp heterozygous 15q13.2q13.3 deletion that was concordant with CMA; however, by leveraging the individual DNA molecule lengths, de novo assembly and haplotype separation, OGM uniquely identified an approximate 2 Mbp inversion in the chromosome without the deletion, along with a 0.5 Mbp deletion compared with the reference (Figure 2B, Sample 15). Both deletions were also identified by the OGM CNV profile. Lastly, OGM identified a 740 kbp tandem duplication on chromosome 16p11.2 (Figure 2C, Sample 38) supported by both pipelines (CNV and assembly). The molecules spanning the fusion point of the duplicated region and its insertion location were assembled into a consensus map demonstrating that the duplication is in tandem.

### 3.4. OGM Resolves the Genomic Structure of CNVs

Out of the 55 studied samples, OGM provided further refinement of the genomic structure in 12 cases (Table 2). The characterization included identification of translocations, determining insertion sites and/or orientation of duplicated regions, and refining the structure of complex rearrangements. In some cases (such as samples 36, 41 and 44), the structure characterized by OGM was consistent with the findings of metaphase FISH visualization; in other cases (such as Samples 28 and 40), OGM was able to characterize further nuances to the structure that FISH was unable to identify.

OGM can identify and define pathogenic unbalanced inter-chromosomal translocations in a single assay without the need for supplemental metaphase FISH testing often performed when CMA identifies CNVs with predicted structural complexity (e.g., a gain and loss at different chromosomal ends). For example, Figure 3A shows a translocation that was natively identified by OGM between chromosomes 2 and 10 with a corresponding copy number gain on chromosome 2 and loss on chromosome 10. Moreover, Figure 3B shows the identification of a putative derivative chromosome 9 as evidenced by copy number gains on chromosomes 9 and 4 with the corresponding fusion between the CNV breakpoint locations.

In addition to the identification of translocations, OGM also identified insertions and in other cases was able to define the underlying genomic structure for duplicated regions (Table 2). In total, OGM identified and refined five insertion locations that were concordant with FISH. For example, a ~150 kbp duplication of chromosome 3 material was inserted into chromosome 17 as defined by FISH and confirmed by OGM (Figure 4A, Sample 28). Notably, using OGM, not only is the insertion location more precisely defined, OGM can also show whether genes are potentially disrupted by the insertion. In this example, the insertion occurs in an area that is duplicated, disrupting *SMG6* or *SGSM2*; however, a normal copy of the two genes still remains due to the duplication within chromosome 17. In a second case, OGM was able to resolve the genomic structure of two neighboring duplications identified by CMA (Figure 4B, Sample 40). OGM CNV results show copy number gains of similar size and in similar locations as CMA; however, consensus genome maps and their corresponding alignments to the reference genome demonstrate the correct genomic structure. As shown in Figure 4B, the two duplications are actually fused, with inversion of the distal duplication. The breakpoints indicate a potential disruption of the *DNMT3A* gene.

Using current standard of care methods, the cases shown in Figure 4 require further manual investigation to interrogate the genomic structure. For example, the 150 kbp chromosome 3 insertion in chromosome 17 is called by the bioinformatics pipeline as an inter-chromosomal translocation; however, this region has two map alignments on each side of the insertion independently aligned to chromosome 17 and none to chromosome 3, confirming an insertion event. The manual shifting of chromosome 17 alignments and collapsing of two independent maps into a single map reveals the structure seen in Figure 4A. Similarly, the two duplications identified in Figure 4B had two supporting maps aligning to each side of the inverted duplicated region (Figure 4B, Ref Chr 2 (C–D interval)). Both of these cases required the manual review of individual single molecules used to construct the contigs. The molecules containing labels spanning into the adjacent maps with correct label alignment demonstrated that the two maps were in fact part of a longer consensus assembly that producing the genomic structures seen in Figure 4.

### 3.5. OGM Filtration Criteria Used to Select for Potential Pathogenic SVs

The initial data analysis was performed in a blinded fashion with a set of OGM specific SV/CNV filtration criteria that masked SV calls, which overlapped areas of the reference genome containing assembly errors, gaps, and lack of sequence at centromeres and telomeres. This filtering step eliminated some OGM SV calls reported by CMA. The filtration criteria were intended to decrease the overall number of SVs requiring manual review while maintaining high sensitivity for potential pathogenic variants. For example, the deletion in Figure 2B was initially filtered out due to flanking LCRs that resulted in a falsely high population frequency calculated at approximately 15%. The overestimation of population frequency of SVs predominantly occurs in low complexity, highly repetitive regions due to misalignments and/or assembly errors. This issue was mitigated by the utilization of the CNV pipeline that called the deletion. Additionally, the filtration criteria were modified to include masked variants that are greater than 1 Mbp in size and a composite clinically relevant gene list was used to enrich for gene-overlapping SVs. Lastly, during the project, a new set of bioinformatics tools were released which allowed for identification of mosaic Y loss, triploidy and AOH (Figure 1).

Using these standardized filtration criteria, we were able to identify copy number gains, losses, inversions, and translocations. The filtering protocol resulted in a considerable decrease in the number of SVs needing expert curation (from 4338 to 17 SVs, per sample, on average (Appendix A)). These variants were then compared to the reported CMA findings. The filtration criteria were able to identify the reported variant/s in 49/51 cases, excluding triploidy and AOH cases. Case 44 was one example of a variant call that was filtered out using the initial criteria, where OGM did not identify the pathogenic variant reported by CMA due to masking of the molecule copy number profile in the telomeric region of chromosome 9. However, a manual review of the region showed a decrease in the copy number to approximately 1× fraction (Appendix A). A second example was Case 12, in which the identified SVs did not overlap the curated disease-causing genes contained in a provided BED file. This indicates that the user specified gene list with corresponding reference coordinates plays an important role in the identification of potential clinically significant variants.

The discrepancy of the SV class in one case demonstrates that the discrepancy often results from regions flanked by LCRs (Figure 2B and Appendix A). It is important to note that these regions suffer from low density probes on all CMA platforms. Also, the reference human genome assemblies have many problematic regions, where the quality of sequence in the centromeric and telomeric regions is less than ideal and can be a problem for most molecular techniques. OGM addresses this issue by potentially masking of the problematic regions, but caution is required when unmasking is needed to assess a true call (Appendix A).

## 4. Discussion

CMA has been implemented globally since it became the first-tier assay for diagnostic evaluation of constitutional disorders. However, the diagnostic success rate of CMA is dependent on the platform and probe coverage and can range from 15–20% [2]. Laboratories that have adopted CMA often explore alternative methods/assays if CMA results are negative. This adds significant cost burden and time to the identification and detection of genomic aberrations in research laboratories and undue burden on patients and families enrolled in research. Optical genome mapping provides a unique and simple workflow and a fast turnaround time to results. OGM specifically allows for the identification of all classes of unbalanced structural variations including those detected by CMA and has the added ability to detect balanced genomic aberrations such as translocations, inversions, and insertions that are cryptic to CMA [9,16,17,20]. Balanced genomic aberrations such as translocations and inversions can cause constitutional disorders through breaking genes, disrupting the regulatory structure or creating gene fusions, in addition, they increase the possibility of having an offspring with an unbalanced genomic rearrangement and raise concern from a reproductive health perspective [23]. In this study, 55 samples harboring 61 reported abnormalities from prior testing (pathogenic, likely pathogenic, or variant of unknown significance) were assessed. OGM results demonstrated 98% concordance with CMA for these variants, with the exception of a case where CMA identified a variant as a duplication and OGM resolved it as an insertion. Unlike CMA, OGM data were also able to provide structural information for CNVs (Table 2). We also investigated the ability of OGM to identify triploidy, UPD and AOH as proof of concept on several samples; however, additional testing is needed, particularly for AOH. Lastly, unlike CMA, OGM identified many more SVs, including balanced events and smaller deletions/duplications (≥500 bp). Since OGM often results in more than four thousand SVs per sample (Appendix A) that need to be effectively filtered to identify clinically relevant and reportable variants. Using this dataset, we established comprehensive filtration criteria for the prioritization of SVs that may be disease associated. It is important to clarify that not all CMA results are equivalent since many different platforms with varying levels of resolution are used globally. Also, the cost of the assay is dependent on the probe density and coverage of the CMA platform. On the other hand, OGM, as a universally adoptable technology, does not depend on any probes or hybridization processes and hence would provide uniform genome-wide coverage at a single price.

### 4.1. Strengths and Limitations of OGM

Of the 61 clinically significant structural variants present in 55 samples, 60 were reproduced by OGM, providing 98% concordance between CMA and OGM (Table 1). Additionally, OGM detected all of the 46 reported pathogenic variants (in 36 samples). Not only does this study demonstrate the high concordance between CMA and OGM, but it is also clear that OGM adds significant value in providing critical and actionable structural information for balanced and unbalanced variants (that CMA is unable to detect). Additionally, OGM was able to better refine the structure of genomes in 12 cases compared with CMA alone (Table 2). Importantly, unbalanced derivative chromosomes are inferred with CMA in three cases, while OGM could confidently detect the translocation (fusion) in all of these cases. This additional information is valuable for clincial research since unbalanced structural rearrangements in affected individuals may be inherited from balanced parental carriers, which adds significant reproductive risks Furthermore, because OGM can identify both cryptic and balanced structural rearrangements, OGM can be an important technology in the cytogenomics lab for elucidating and resolving the potential genetic causes leading to specific disease phenotypes.

As with any other molecular methodology, OGM has technical limitations. Both CMA and OGM are unable to resolve balanced Robertsonian translocations due to the repetitive nature of the centromeric or the p-arm breakpoint regions of acrocentric chromosomes (Appendix A). For Case 46 in this study, metaphase FISH performed after CMA identified the unbalanced Robertsonian translocation. Similarly, both OGM and CMA can identify the presence of copy number gains near centromeres, which may be associated with marker chromosomes, but neither can conclusively identify supernumerary marker chromosomes because of the lack of reliable coverage of pericentromeric and centromeric DNA (Appendix A, Sample 32). Lastly, although CNVs are called in the telomeric regions by OGM, the individual molecule alignments are often noisy due to repetitive DNA sequences, inaccurate reference assembly, and in some cases lack of the specific sequences for OGM labeling. This can lead to masking of SV calls, even if the CNV pipeline demonstrates the copy number change (Appendix A, Sample 44). Telomeric fusions may also reduce the efficiency of OGM to identify an insertion location for some copy number gains as evident in case 33, where a duplication of distal 12q is inserted at the 12q terminus in an inverted orientation as identified by FISH (Appendix A); however, manual investigation of OGM maps and molecules suggested a possible insertion location and inverted orientation. A similar mechanism can be observed for case 39.

Another case of interest highlighting differences between OGM, and CMA is sample 27. CMA identified a small duplication (44 kbp, arr[GRCh37] 2q35(219890098_219934462)x3) involving the *IHH* gene. OGM did not identify the same duplication involving *IHH*, instead it made a similarly sized 45 kbp (ogm[GRCh37] 2q35(219823156_219844642)x3) insertion call adjacent to the *IHH* gene (Appendix A). OGM was unable to disambiguate the content of the inserted region due to the presence of only two labels that could not be accurately mapped. However, the insertion location of the duplication identified by CMA could be mapped by OGM to a location ~45.5 kbp upstream of *IHH* and was determined to not disrupt the original copy of the *IHH* gene. The accurate knowledge of what is duplicated as well as the insertion locations of duplicated regions provide valuable information that can be used for interpretation of the detected variants. This example demonstrates the benefits and limitations of both CMA and OGM technologies.

### 4.2. CNV Size Discrepancies between CMA and OGM

The underlying technology for determining CNV sizes between CMA and OGM are different, which directly translates into a predictable discrepancy in the size of calls made between the two technologies. Hybridization of oligonucleotide and SNP probes in CMA targeted throughout the human genome results in uneven coverage, leading to CNV sizing limitations, particularly in intergenic and repetitive regions. Sizing is dependent on the proximity of unaffected probes to the putative CNV breakpoints (i.e., CMA reports the minimum size for breakpoint locations of identified CNVs). In contrast, OGM relies on utilization of many long molecules for genome assembly with the ability to accurately measure the length of DNA at any given region of the genome (i.e., assembled map information provides accurate SV sizing within approximately 60 bp). However, the calling of breakpoint locations are dependent on label density and in contrast to CMA, OGM calls the largest possible coordinates (+/−3.3 kbp).

OGM leverages a second and complementary method for CNV calling based on counting the depth of coverage of the mapped molecules in the genome thereby enabling a confirmatory measurement for larger CNVs (i.e., read depth CNV calling starts at 500 kbp). This depth of coverage method also complements the determination and assessment of numerical whole-chromosome aneuploidies such as monosomies and trisomies. Finally, it complements assembly-based calling for certain CNVs like centromeric unbalanced translocations and Robertsonian translocations. The precision of this read-depth-based CNV calling is lower than assembly-based CNV calling, so the assembly-based method should always be prioritized when both algorithms make concordant calls.

For the past decade, CMA has been used globally by the cytogenetics community for clinical diagnostic, research and translational use. Newer sequencing-based technologies profess the ability to detect CNVs; however, the specificity and sensitivity is dependent on the genome content and depth of coverage of the sequencing platforms [24]. Whole-genome sequencing is also being evaluated for SV detection, but is cost prohibitive for global adoption [25,26]. OGM demonstrates a unique ability to fit into any cytogenetic workflow and allows the clinical research community to benefit from not only detecting the SVs and also the genomic architecture underlying the SV formation. The detection of these SVs provides unique value to clinical researchers treating individuals and families affected with a genetic disorder. Accurate detection of genomic aberrations is important for appropriate management/intervention and to provide relevant information for appropriate genetic counseling as part of reproductive health. This study was performed on a samples harboring variety of different classes of SVs demonstrates that the CNV sizes between CMA and OGM were concordant and is in line with other published literature [17,27].

## 5. Conclusions

In this study, the technical concordance and analytical validity of OGM was compared to CMA in a cohort of well-characterized samples with both numerical and structural anomalies. OGM achieved 98% concordance with CNVs identified by CMA, but also aided in the better refinement of the genomic architecture surrounding several CNVs. Since CMA can only detect CNVs and cannot discern the nature of most structural variants, supplementary techniques like FISH or karyotypingare often added for a more comprehensive assessment of the variant. In a single assay, OGM can accurately identify both balanced and unbalanced SVs, triploidy, and large AOHs, thereby fully resolving many variants without the need for additional methods. Taken together, these results and other studies show high concordance of OGM with multiple cytogenetic methods [17] and increased ability to detect pathogenic findings [28]. A recent large-cohort blinded study of retrospective cases including individuals suspected of having a genetic condition and having received a previous genetic test showed that increasing the burden of variant interpretation due to higher detection rate of OGM can be alleviated by systematic filtering and utilization of genetic databases, thereby reducing the effort required for a case review [27]. This study adds to the growing body of evidence supporting the implementation of OGM as a first-tier testing method that provides comprehensive results in a cost-effective and [27,29,30,31] streamlined workflow.

## Figures and Tables

**Figure 1 genes-14-01868-f001:**
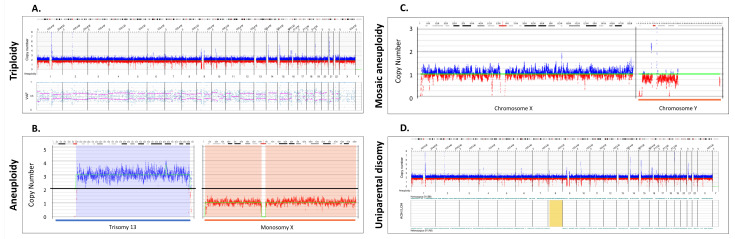
Whole chromosomal abnormalities and absence of heterozygosity. (**A**) A triploid genome (69,XXX, Sample 51) showing CN profile and variant allele fraction profiles (VAF). OGM software (Bionano Access v.1.7) automatically quantifies VAF for all variants and constructs a plot depicting the genome wide distributions, shown in the bottom part of (**A**). In cases of a triplication the VAF are distributed differently compared to diploid chromosomes: VAF around 1 for variants present in 3 alleles, 0.67 for variants present in 2 alleles, and 0.33 for variants present in only 1 allele (VAF around 0.67 and 0.33 indicated by pink lines, see also Appendix A). (**B**) Copy number profile displaying two aneuploidies: trisomy 13 (Sample 48) and monosomy X (Sample 49). The Y axis represents the copy number measurement with the black line centered at two copies. Blue lines above the baseline represent gains and red losses. The cytobands for each of the chromosomes are displayed on the top. (**C**) Copy number profile displaying a mosaic loss of the Y chromosome (Sample 47). (**D**) AOH and CNV profiles displaying regions on chromosome 8 that do not have heterozygous variants indicating a potential uniparental disomy, highlighted in yellow (Sample 52).

**Figure 2 genes-14-01868-f002:**
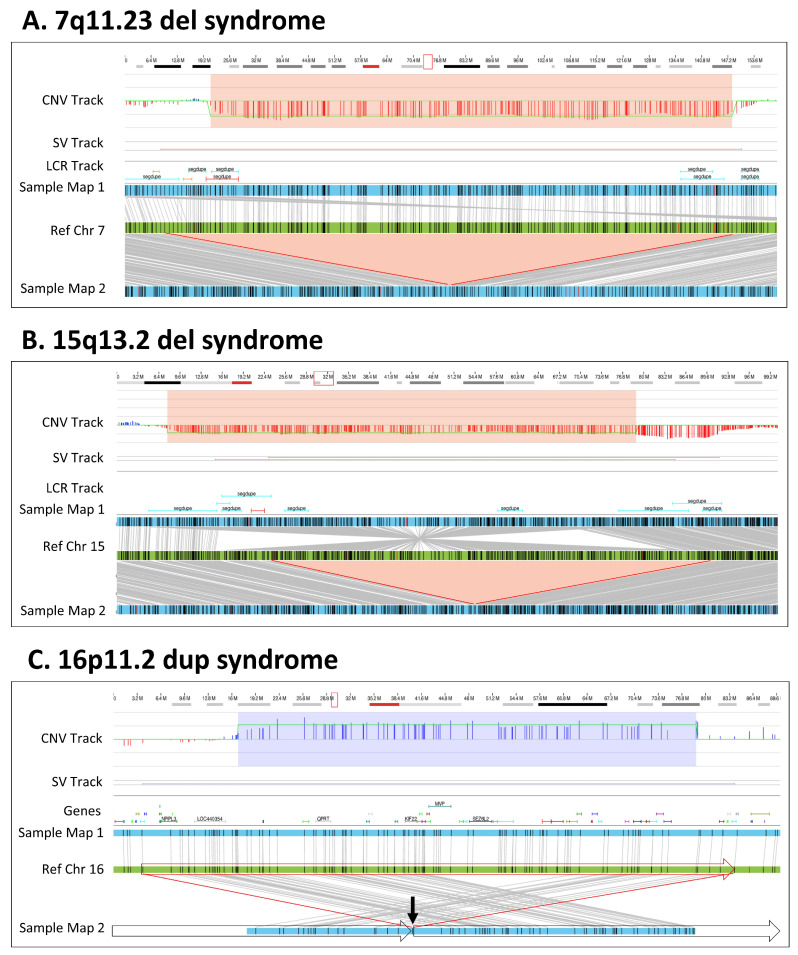
Microdeletion/duplication syndromes. (**A**) A 1.9 Mbp heterozygous copy number loss in the 7q11.23 region. The red box in the cytoband on the top of the figure indicates the region of interest that is shown below. The deletion is captured by two OGM variant-calling algorithms—the copy number and the de novo assembly algorithms. In the top track, the copy number profile shows a one-copy drop. The bottom track shows that two assembled maps in blue align to the reference in green. The upper assembled Map 1 represents the reference allele, whereas the lower Map 2 captures the 1.9 Mbp deletion. Together the maps indicate that the deletion is heterozygous (Sample 8). (**B**) A 1.9 Mbp heterozygous copy number loss in the 15q13 region. The top track shows that the deletion is called by the copy number algorithm. The assembly pipeline shows that two distinct haplotype resolved alleles; one precisely shows the 1.9 Mbp deletion (Map 2) and the other (Map 1) carries an inversion with an additional 0.5 Mbp loss compared with the reference (Sample 15). (**C**) A 0.7 Mbp tandem duplication in 16p11.2. The copy number profile indicates a copy number of three. The de novo assembly delineates the structure and orientation of the duplication; the three copies occur on two haplotypes, with one copy on Map 1 and two copies in tandem order on Map 2. Due to the size of the duplication, the OGM molecules do not cover the entirety of the duplication. Instead, the map alignments show the head-to-tail fusion point indicated by the arrow and subsequent alignments on either side of the duplication. The genomic structure is shown with the boxed arrows around the sample Map 2 (Sample 38).

**Figure 3 genes-14-01868-f003:**
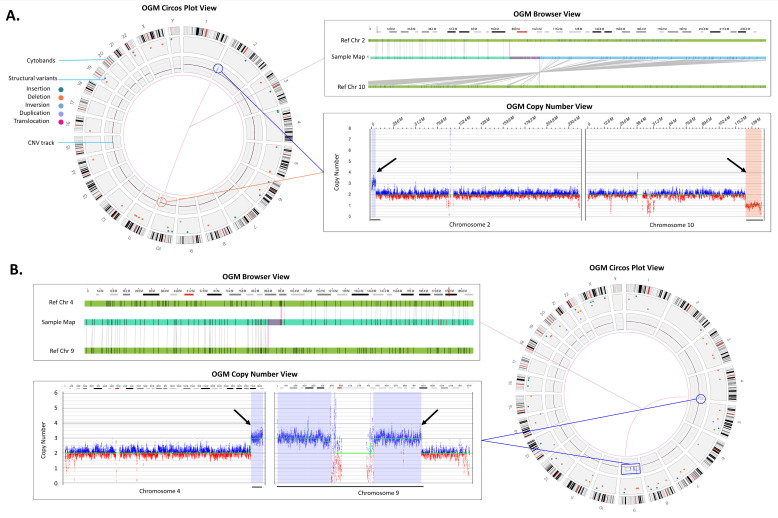
Translocations. (**A**) An unbalanced translocation detected between chromosomes 2 and 10. Left panel: Circos plot summary displaying SVs unique compared to the Bionano control database, sample (Sample 36). The translocation and the accompanying gain in 2p (blue line and circle) and loss in 10q (red line and circle) are shown via a line connection. Top right panel: The red box in the cytoband on the top indicates the region of interest that is shown below. The genome browser view details the alignment of the sample’s consensus map (light blue bar) with the reference consensus maps (light green bars) and provides the detail of the structural variation. Bottom right panel: The *Y*-axis represents the copy number level and *X*-axis gives the chromosome position, the CNV plot showing gain on chromosome 2 and loss on chromosome 10 (black arrows). (**B**) Rearrangements indicating the presence of a derivative chromosome (Sample 41). Top left panel shows a zoomed in view of a t(4;9) translocation. Bottom left panel shows copy number gains whose breakpoints coincide with the translocation breakpoints (black arrows). Combining both events (blue line and circle) in the circos plot on the right panel, we can infer that the gains and fusions between chromosomes 4 and 9 represent +der(9)t(4;9).

**Figure 4 genes-14-01868-f004:**
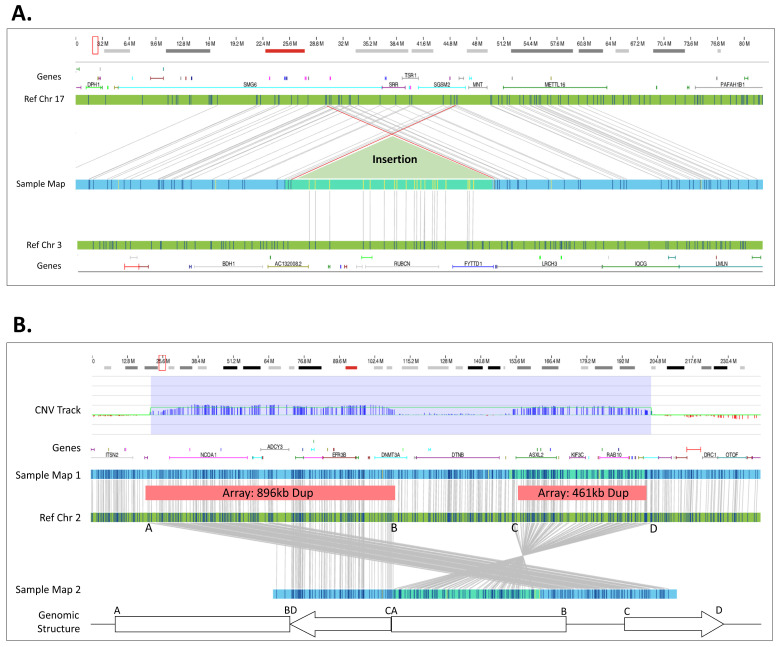
Insertion and complex structure. (**A**) The red box in the cytoband on the top of the figure indicates the region of interest that is shown below. A 151.3 kbp segment of 3q29 was duplicated and reinserted into 17p13.3. However, the insertion site shows additional complexity where 120.2 kbp of 17p13.3 around the insertion site is also duplicated (Sample 28). (**B**) Two duplications 896.3 kbp and a 461.2 kbp occur in proximity. The CNV track shows that the copy number algorithm detected the two duplications. The assembly assembled two different haplotypes: one with and one without the duplications. Based on the structure of Map 2, we deduce the duplications structure as depicted in the bottom (Sample 40).

**Table 1 genes-14-01868-t001:** Concordance of CNVs between OGM and CMA.

Method	Copy Number Variants				Total
	Aneuploidy	Triploidy	Gain	Loss	
CMA	5	2	21	33	61
OGM	5	2	20 *	33	60
Concordance	100%	100%	95%	100%	98%

* CMA detected a copy number gain, but OGM identified the similarly sized insertion (Appendix A, Sample 27).

**Table 2 genes-14-01868-t002:** Better characterization and refinement of the genomic structure by OGM.

Sample	CMA	FISH	OGM	Result
36	arr[GRCh37] 2p25.3p25.2(36400_4801965)x3,10q26.12q26.3(123027554_135403394)x1	der(10)t(2;10)(p25.2;q26.12)	ogm[GRCh37] t(2;10)(p25.2;q26.1)(4776157;123014483),2p25.3p25.2(15924_4746589)x3,10q26.12q26.3(123011875_135522591)x1	OGM identified CNVs and translocation.
41	arr[GRCh37] 4q34.3q35.2(179412576_190896674)x3,9p24.3q31.1(209020_105724992)x3	+der(9)t(4;9)(q34.3;q31.1)	ogm[GRCh37] t(4;9)(q34.3;q31.1)(179395177;105721182),4q34.3q35.2(179395177_191040751)x3,9p24.3q31.1(14556_105718660)x3,	OGM identified CNVs and translocation.
44	arr[GRCh37] 1q21.1q21.2(146531538_147726541)x3,9q34.3(139610281_141005513)x1,21q22.13q22.3(38319773_48091215)x3	der(9)t(9;21)(q34.3;q22.13),(1q21.1)x3	ogm[GRCh37] t(9;21)(q34.3;q22.13)(136684084;36948511),1q21.1q21.2(146057345_148928812)x3,9q34.3(136472755_138334464)x1,21q22.13q22.3(36948511_46697230)x3	OGM identified CNVs and translocation.
26	arr[GRCh37] 2p11.2(85233220_85575202)x3	der(2)ins(2)(p?15;p11.2p11.2)	ogm[GRCh37] ins(2;2)(p16.1;p11.2)(57631805;85217155_85572976)	OGM defined CNV insertion site.
28	arr[GRCh37] 3q29(197398764_197495350)x3	der(17)ins(17;3)(p13.?3;q29q29)	ogm[GRCh37] ins(17;3)(p13.3;q29)(2155812/2276044;197344236_197495529)	OGM defined CNV insertion site.
30	arr[GRCh37] 6q14.1(77294196_77479434)x3	der(X)ins(X;6)(Xq2?8;q14.1q14.1)	ogm[GRCh37] ins(X;6)(q28;q14.1)(152467195;77273466_77477943)	OGM defined CNV insertion site.
31	arr[GRCh37] 6q14.1(77294196_77479434)x3	der(X)ins(X;6)(Xq2?8;q14.1q14.1)	ogm[GRCh37] ins(X;6)(q28;q14.1)(152467195;77273466_77477943)	OGM defined CNV insertion site.
24	arr[GRCh37] 1p36.22(11517159_11892978)x3	(1p36.22)x3	ogm[GRCh37] 1p36.22(11518058_11896987)x3	Tandem duplication (defining orientation)
38	arr[GRCh37] 16p11.2(29657192_30192346)x3	(16p11.2)x3	ogm[GRCh37] 16p11.2(29463027_30202372)x3	Tandem duplication (defining orientation)
35	arr[GRCh37] Xp21.3p21.2(28916857_29457146)x2	(Xp21.3)x3	ogm[GRCh37] Xp21.3(28912348_29459420)x2	Tandem duplication (defining orientation)
28	arr[GRCh37] 6q12(76287632_77298392)x3	(6q14.1)x3	ogm[GRCh37] 6q12(76222934_77313055)x3	Tandem duplication (defining orientation)
40	arr[GRCh37] 2p23.3(24633371_25529639)x3,2p23.3(25961533_26422725)x3	(2p23.3)x3	ogm[GRCh37] 2p23.3(24643804_25517100)x3,2q23.3(25943863_26441430)x3	OGM identified both CNVs and characterized their structure.

Summary of variants identified by CMA for which OGM further refined the genomic structure. Only SVs for which OGM refined the genomic structure compared with CMA are listed in the corresponding columns.

## Data Availability

All data supporting this study is included with the paper with the exception of individual alignment and variant call files, to comply with Health Insurance Portability and Accountability Act of 1996 (HIPAA) protections and the consent for aggregate, de-identified research approved IRB protocol (IRB-20216077).

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
