# Peer review of "Comparative Benchmarking of Optical Genome Mapping and Chromosomal Microarray Reveals High Technological Concordance in CNV Identification and Additional Structural Variant Refinement"

_genes, 2023, doi:10.3390/genes14101868_

Round 1

Reviewer 1 Report

This study compares the efficacy of OGM to CMA in identifying clinically reported genomic variants, and concluded a comparable performance of OGM but higher potential for clinical utility of OGM when comparing to CMA. The manuscript is well-written and the scientific logical is easy to follow, but I still have some questions regarding to their presented results.

1.  In Figure 2C, due to that the duplication size was not fully covered, how did authors make the conclusion of a tandom duplication instead of a chromosome rearrangement based on the alignment results in Sample 38?

2.  The authors listed and summarized the observations on genomic structures that can be refined by OGM comparing to CMA. How significant would OGM outperform CMA in this regards? It would be good for authors to show the summary of the numbers of chromosome relocation events that can be implicated by CMA but not by OGM, as well as those can be inferred  by both approaches. 

Reviewer 2 Report

 Barseghyan and co-authors performed a comparison study between chromosome microarray (CMA) using 180K SNP Agilent array and Optical Genome Mapping (OGM) to investigate the concordance and detection sensitivity. The study is described in detail with adequate examples and illustrations. They find 100% concordance for clinical reportable CNVs and 98% overall concordance which is very high. In addition, OGM allows identification of balanced structural variants which is important to elucidate the complexity of rearrangements and identify potential carriers of balanced genomic anomalies.

 Comments:

The authors propose OGM as a first-tier analysis in DD/ID genetic investigation. However, the data provided here is somewhat optimistic to come to this conclusion, because cohort of patient in whom clinically significant CNVs were already identified by CMA were tested with OGM. Since OGM provides more information about structural balanced variants a larger data set would be needed including patient in which no causal SV or CNV is identified. Although the filtering in the software is explained OGM approach will unavoidably also identify rare benign SVs, which will be difficult to interpret their clinical significance. To implement OGM as first-tier test in a routine work-up may cause overinterpretation of benign SVs. A comment regarding interpretation of rare SVs should be added in this context.

Reviewer 3 Report

Thank you to the authors for this well-written study, which has contributed to our understanding of current genomic technologies. I have a few comments regarding this manuscript.

My main concern pertains to the advantage of OGM over CMA in detecting balanced translocations. It is widely recognized that CMA cannot detect balanced translocations. On the other hand, OGM claims the capability to identify various types of structural variations, including balanced translocations. While this manuscript effectively illustrates how OGM can detect unbalanced translocations, deletions/duplications, insertions, and other SVs, it lacks an example case demonstrating the detection of a balanced translocation. To make it more convincing, I suggest including a case where OGM successfully identified a chromosomal detected balanced translocation that CMA failed to detect.

Furthermore, I recommend expanding the discussion section to cover the significance of balanced translocation detection by OGM. This is a crucial advantage that OGM holds over CMA, and it would be beneficial to elaborate on its clinical relevance and implications.

In addition, I've noticed a few minor points in Table 2 that require attention:

1. Kindly review the OGM ISCN for the insertion cases 28, 30, and 31. The ISCN appears to indicate only insertions, with duplications not being displayed. While I am not familiar with OGM ISCN, I recommend double-checking to ensure its accuracy.

2. Case 40: There is a typographical error in the second 2p duplication. The CMA result specifies 2p23.3, whereas OGM indicates 2p24.3. Please correct this discrepancy for accuracy.
